# Is Antimicrobial Dosing Adjustment Associated with Better Outcomes in Patients with Severe Obesity and Bloodstream Infections? An Exploratory Study

**DOI:** 10.3390/antibiotics9100707

**Published:** 2020-10-16

**Authors:** Stéphanie Sirard, Claire Nour Abou Chakra, Marie-France Langlois, Julie Perron, Alex Carignan, Louis Valiquette

**Affiliations:** 1Department of Microbiology and Infectious Diseases, Université de Sherbrooke, Sherbrooke, QC J1H 5N4, Canada; stephanie.sirard@USherbrooke.ca (S.S.); claire.nour.abou.chakra@usherbrooke.ca (C.N.A.C.); alex.carignan@usherbrooke.ca (A.C.); 2Department of Medicine, Division of Endocrinology, Université de Sherbrooke, Sherbrooke, QC J1H 5N4, Canada; marie-france.langlois@usherbrooke.ca; 3Department of Pharmacy, Centre Intégré Universitaire de Santé et de Services Sociaux de l’Estrie-Centre Hospitalier Universitaire de Sherbrooke, Granby, QC J2G 1T7, Canada; julie.perron2@usherbrooke.ca

**Keywords:** obesity, bloodstream infection, antimicrobials, prescription

## Abstract

The impact of adjusted treatment on clinical outcomes in patients with severe obesity is unclear. This study included adults with severe obesity admitted for bloodstream infections between 2005 and 2015. The patients were grouped according to the percentage of the appropriateness of the dosage of their antimicrobial treatment: 80–100% = good, 20–79% = moderate, and 0–19% = poor. The association between antimicrobial adjustment and a composite of unfavourable outcomes [intensive care unit stay ≥72 h, duration of sepsis >3 days, length of stay ≥7 days or all-cause 30-day mortality] was assessed using logistic regression. Of 110 included episodes, the adjustment was rated good in 47 (43%) episodes, moderate in 31 (28%), and poor in 32 (29%). Older age, Pitt bacteremia score ≥2, sepsis on day 1, and infection site were independent risk factors for unfavourable outcomes. The level of appropriateness was not associated with unfavourable outcomes. The number of antimicrobials, consultation with an infectious disease specialist, blood urea nitrogen 7–10.9 mmol/L, and hemodialysis were significantly associated with adjusted antimicrobial dosing. While the severity of the infection had a substantial impact on the measured outcomes, we did not find an association between dosing optimization and better outcomes.

## 1. Introduction

In the last 40 years, the prevalence of obesity has doubled in more than 70 countries, accounting for over 2 million deaths worldwide [1]. Aside from comorbidities such as type 2 diabetes, hypertension, and cardiovascular diseases, obesity is associated with a high risk of infections [2,3]. Physiologic alterations in patients with obesity influence the pharmacokinetics (PK) and pharmacodynamics (PD) of many drugs, including antimicrobials [4,5]. Underdosing of antimicrobials in patients with obesity could lead to sub-inhibitory concentrations, which, in turn, could impair treatment and lead to clinical failure [6,7,8,9]. For example, in patients with severe obesity and cellulitis, a low antimicrobial dose upon hospital discharge was associated with either recurrence, emergency room visit, rehospitalization, or 30-day attributable death (odds ratio [OR] 3.6 95% CI 1.4–9.4) [7]. In a cohort of critically ill patients with complicated intra-abdominal and skin and soft tissue infections, high doses of tigecycline resulted in a significant reduction in mortality, intensive care unit (ICU) length of stay (LOS), and occurrence of bacteremia and septic shock [10].

Bloodstream infections (BSI) are severe infections and one of the leading causes of death in North America and Europe [11]. In one study, the risk of BSI was higher in patients with obesity than in normal-weight patients (31% for body mass index (BMI) of 30–34.9, 87% for BMI of 35–39.9, and 210% for BMI of ≥40) [12]. While obesity had no association with short-term all-cause mortality and clinical outcomes in patients with BSIs [13], another study found that high BMI was associated with organ failure and all-cause hospital mortality in patients with BSIs due to Gram-negative bacteria [14].

Although some studies have investigated the link between obesity and unfavourable outcomes associated with BSI, to our knowledge, none has focused on the impact of antimicrobial dose adjustment in BSI patients with class III obesity [12,13,14,15,16].

In this article, we describe a retrospective cohort of adults with class III obesity hospitalized for BSI, where we assessed factors associated with adjustment of antimicrobial dosing and compared clinical outcomes according to the appropriateness of antimicrobial dose adjustments.

## 2. Materials and Methods

### 2.1. Population and Study Design

This retrospective study was conducted at the Centre intégré universitaire de santé et de services sociaux de l’Estrie-Centre hospitalier universitaire de Sherbrooke (CIUSSSE-CHUS), a 677-bed academic centre in the Province of Quebec, Canada. Approval was obtained from CIUSSSE-CHUS institutional review board (#12–187). Subjects were identified through a clinical data warehouse. All adult patients with documented class III obesity (BMI ≥ 40 kg/m^2^) hospitalized between 1 August 2005 and 31 August 2015 for BSI were included.

BSI was defined by the presence of a pathogen in one blood culture or the presence of a skin flora microorganism (coagulase-negative staphylococci, alpha-hemolytic streptococci, *Micrococcus* species, *Propionibacterium*/*Cutibacterium* species, *Corynebacterium* species, and *Bacillus* species) in at least two consecutive blood cultures (from two different sites).

Specific populations for whom the BMI was not reliable, such as pregnant women, patients with dwarfism, those with above-the-knee bilateral amputation, or those with a history of bariatric surgery were excluded. Other exclusion criteria were: presence of fungemia, transfer from another hospital after ≥48 h, receiving palliative care, presence of more than one bacterial infection, or two or more distinct episodes during hospitalization. We excluded patients who were treated for the whole or the majority of the treatment (>80%) with an antimicrobial requiring no adjustment for obesity (cefixime, moxifloxacin, ertapenem, fosfomycin, and tigecycline) and those treated with vancomycin or aminoglycosides only, especially cases where it was the only effective antimicrobial. In addition, patients who died or did not received an antimicrobial within the first 48 h after the initial positive blood culture, or had inadequate antimicrobial coverage for >48 h were excluded. 

### 2.2. Data Collection

A standardized form was used to collect data on clinical variables from computerized medical charts. Pathogens isolated from blood samples were noted along with antimicrobial susceptibility test results. Immunosuppression was defined as the presence of leukaemia, lymphoma, HIV infection, neutropenia (neutrophils < 1800/µL), organ transplantation, and connective tissue disease or use of immunosuppressive drugs for over one month within the previous six months. To evaluate the severity of illness, the Pitt bacteremia score (PBS) (ranging from 0 to 18) was documented on the day of the positive blood culture and up to 48 h prior [17,18]. The time to effective antimicrobial was determined by the time between admission and the administration of the first effective antimicrobial related to the infection. All antimicrobial prescriptions relevant to the treatment of BSI were reviewed for the route of administration, dose, and dosing interval and were compared to the local guidelines for adults with class III obesity based on the current literature (see Table 1) [5,19,20].

Cockcroft-Gault equation with adjusted body weight was used to estimate creatinine clearance (CrCl) at the beginning of all prescriptions and for every significant change in creatinine values [21]. A prescription was deemed inadequate if either the dose and/or the dosing intervals were lower than expected for class III obesity. When multiple antimicrobials were administered at the same time, we considered the whole prescription adequate if at least one antimicrobial dosing and the spectrum were adequate. The first prescription was also carefully reviewed. The percentage of the appropriateness of the dose and dosing intervals was calculated by dividing the number of days of adequate treatment by the total number of days of treatment and was considered good (80–100%), moderate (20–79%), or poor (0–19%). This classification has been selected after discussion with infectious disease experts, locally. 

### 2.3. Outcomes

To reflect unfavourable outcomes potentially associated with unadjusted antimicrobial dosing and because of the low frequency of each component, we constructed a composite primary endpoint including clinically relevant components: ICU stay ≥72 h, duration of sepsis >3 days, LOS ≥7 days, or all-cause 30-day mortality. Other secondary endpoints collected per hospitalization were time to defervescence, time to white blood cells (WBC) normalization, time to sepsis normalization, ICU LOS, need for and duration of mechanical ventilation, and hospital LOS. Fever was defined as an increase in body temperature above 37.5 °C orally, 38 °C rectally and centrally, or 37.3 °C axillary. Time to defervescence was the time between the first abnormal value and the first normal value that remained within normal values for at least 48 h. Only the first febrile episode within 48 h of the first positive blood culture was considered in this calculation. WBC normalization associated with the first positive blood culture was defined as a stable return within the normal range during hospitalization. We could not calculate this variable in patients for whom WBC counts were within or below normal range during the entire study period. To define sepsis, we used a modified sequential organ failure assessment (mSOFA) to overcome the limitations due to missing values [22,23]. A serial mSOFA was calculated on days 1, 3, 5, and 7 with the most abnormal values in the 24-h period. Missing values were imputed with normal values, and the corresponding parameter of mSOFA was attributed a score of 0. Sepsis was defined as an mSOFA score of ≥2. All-cause readmission within 30 days of hospital discharge was assessed, and relapse was considered when patients were hospitalized for the initial infection or a complication. 

### 2.4. Statistical Analysis

Data were analyzed using IBM SPSS Statistics for Windows, version 25 (IBM Corp., Armonk, NY, USA). Groups of patients were compared on the basis of the appropriateness of the antimicrobial dosage. To account for potential changes linked to the impact of an antimicrobial stewardship program based on a decision support system (APSS, Lumed Inc., Canada) implemented in August 2010, we divided the study period into three segments: pre-APSS (2005–July 2010), early-APSS (August 2010–2012), and late-APSS (2013–2015). The Antimicrobial Prescription Surveillance System (APSS) is an asynchronous system that generates alerts for potentially inappropriate antimicrobial prescriptions based on published recommendations and expert opinions. These alerts are reviewed by pharmacists who are part of the antimicrobial stewardship program team and recommendations are made to physicians. Special rules were developed for patients with class III obesity [24].

Descriptive statistics were used to characterize baseline demographic characteristics, comorbidities, and outcomes, stratified by the level of appropriateness. Descriptive analyses are presented using three groups based on the level of appropriateness (good, moderate and poor). For some comparisons, we combined the moderate and poor groups and compared to the most optimal group (good). Categorical variables were reported as number and percentage for each group and were compared using the χ^2^ test or binary logistic regression, when appropriate. Continuous variables were reported as median values with their interquartile range (IQR) and were compared with the Wilcoxon test. Logistic regression was used to assess the association between adjusted antimicrobial dosing and unfavourable outcomes and to identify factors associated with adjusted antimicrobial therapy (0–19% poor compared to 20–100%). Selected variables and variables identified in univariable analysis were included in a multivariable model in order of the lowest *P*-value and results of the likelihood ratio test. The results are presented as unadjusted or adjusted OR (aOR) with 95% confidence interval (CI).

## 3. Results

During the study period, 160 clinical episodes of positive blood cultures in adults with class III obesity were identified in our centre, and 110 episodes occurring in 96 patients met the eligibility criteria (Appendix A). The excluded patients were similar to the study population, except for higher rates of intra-abdominal (16% vs. 5%, *p* = 0.04) and catheter (16% vs. 4%, *p* = 0.009) infections. Patients’ characteristics and comorbidities are presented in Table 2. Antimicrobial treatment was classified as 80–100% adequate (good) in 47, 20–79% adequate (moderate) in 31, and 0–19% adequate (poor) in 32 patients.

Overall, the median BMI was 44.9 kg/m^2^ (IQR 42–49), 20% (*n* = 22) of the patients had a BMI over 50 kg/m^2^, and 85% (*n* = 94) had at least one chronic underlying illness. The most frequent comorbidities were diabetes (69%, *n* = 76), coronary artery disease (32%, *n* = 35), and chronic obstructive pulmonary disease (25%, *n* = 27). One in five patients (21%, *n* = 23) had renal failure and 15% (*n* = 17) were immunocompromised. Apart from a significantly greater proportion of hemodialysis patients in the group with good adjustment, all other demographic variables and comorbidities were similar between groups.

The most common source of BSI was urinary tract infections (34%), followed by skin and soft tissue infections (25%). Infections in patients who had good antimicrobial adjustment were more severe, with a lower proportion of urinary tract infections (21% vs. 43%, *p* = 0.018), and a greater proportion of patients with a Pitt bacteremia score (PBS) ≥ 2 (68% vs. 40%, *p* = 0.003); there was a significantly higher frequency of sepsis in this group than in those with moderate or poor levels of adjustment (79% vs. 59%, *p* = 0.027). *Escherichia coli* was the most frequently isolated pathogen (28% of episodes). Enterobacteriaceae were recovered less often from patients with a good adjustment than from patients in the other groups (28% vs. 51%, *p* = 0.015).

During hospitalization, patients received an average of 3.1 ± 1.2 antimicrobials for their infection, of which 1.8 ± 0.9 had inadequate posology. The first prescription was unadjusted for the dose and/or the interval in 60% of patients (*n* = 66), and the dose was insufficient in 68% of the cases. Piperacillin-tazobactam (25%), ciprofloxacin (20%), and ceftriaxone (10%) were the most frequently non-adjusted antimicrobials. 

More than half of the episodes (54%, *n* = 59) occurred after the implementation of APSS. There was a significant increase in the median appropriateness percentage of the treatment in the late-APSS period (84% [IQR 35–100], *p* = 0.031) compared with the other periods (pre-APSS: 27% [IQR 12–86]; early-APSS: 60% [IQR 9–97]) (Suppl. Data, Appendix A). The proportion of inadequate prescriptions upon discharge was significantly lower (44% vs. 75% *p* = 0.02) in the late-APSS than in the pre-APSS period. Further, consultation with an infectious disease specialist was more frequent among patients with a good level of appropriateness than in the other categories (66% vs. 37%, *p* = 0.002).

### 3.1. Outcomes

The clinical outcomes (hospital outcomes and 30-day outcomes) are presented in Table 3. Overall, 53% (*n* = 58) of patients were admitted to the ICU and the median time to ICU admission was 7.4 h (IQR 4.0–14.9). Patients in the good appropriateness category tended to be admitted sooner (5.6 h IQR 3.7–10.4, *p* = 0.25) compared with the other groups (9.4 h [IQR 5.0–25.4]; 8.0 h [IQR 4.0–21.8]). Time from admission to first effective antimicrobial did not differ between groups (*p* = 0.84). The first antimicrobial was administered before ICU admission in most patients (89%, *n* = 98), but half of the patients who received their first antimicrobial in the ICU were in the good level of appropriateness. The patients in this group experienced more sepsis on days three and five and required more mechanical ventilation. 

Although more patients from the good and moderate appropriateness groups were readmitted within 30 days from discharge and had high mortality rates, these differences did not reach statistical significance.

### 3.2. Factors Associated with Adjusted Antimicrobial Therapy

Factors associated with antimicrobial dosing adjusted for obesity are presented in Table 4. In the adjusted model, the number of antimicrobials (aOR 2.2, 95% CI 1.4–3.4,), consultation with an infectious disease specialist (aOR 3.3, 95% CI 1.3–8.6), blood urea nitrogen (BUN) 7–10.9 mmol/L (aOR 7.3, 95% CI 1.8–29.5), and hemodialysis (aOR 10.30, 95% CI 1.62–65.56) were significantly associated with high appropriateness. BMI >50 or weight >120 kg was not associated with adjusted antimicrobial dosing.

### 3.3. Factors Associated with Unfavourable Outcomes

Overall, 55% (*n* = 60) of the patients had at least one of the following components of a composite outcome: ICU stay ≥72 h (33%, *n* = 36), duration of sepsis >3 days (34%, *n* = 37), LOS ≥7 days (55%, *n* = 61), and 30-day mortality (8%, *n* = 9). Risk factors for unfavourable outcomes are shown in Table 5. In multivariable analysis, age (aOR 1.07, 95% CI 1.02–1.12, *p* = 0.009), PBS ≥2 (aOR 7.30, 95% CI 2.09–25.52, *p* = 0.002), sepsis on day 1 (aOR 16.78, 95% CI 3.93–71.63, *p* < 0.001), and infection site (pulmonary aOR 7.52, 95% CI 1.20–47.15, *p* = 0.031, skin and soft tissue aOR 7.79, 95% CI 1.67–36.41, *p* = 0.009, others aOR 9.47, 95% CI 1.99–45.10, *p* = 0.005) were significantly associated with unfavourable outcomes. After adjustment, no measure of treatment appropriateness (first adjusted prescription, adjusted prescription within the first 72 h, and level of appropriateness) was associated with unfavourable outcomes.

## 4. Discussion

Since the prevalence of obesity continues to rise, and as individuals with obesity are likely to receive a high number of antimicrobials [25,26] and complex antimicrobial treatment [27], a better understanding of the impact of optimal dosing adjustment in patients with obesity is needed. In this study, we retrospectively assessed the impact of the appropriateness of antimicrobial dosing in patients with severe obesity hospitalized for BSI. 

We observed low adherence to our local guidelines on the adjustment of doses for patients with severe obesity, with the first prescription being adequate in 40% of the episodes and the treatment being fully adequate in only 24% of the cases. These findings are consistent with those of previous studies, where recommendations (published or local guidelines) were rarely followed [28,29,30,31]. For instance, in patients with class III obesity, initial doses of vancomycin [28], ciprofloxacin, cefazolin, and cefepime [31] were adequate in only 0%, 1.2%, 3%, and 8% of the cases, respectively. However, in our centre, a computerized clinical decision support system designed to assist the antimicrobial stewardship program team [24] had an impact on the prescriptions for patients with obesity, as shown by a three-fold increase in the median appropriateness of the antimicrobial treatment from the pre-APSS to the late-APSS period. In addition, patients who benefited from a consultation in infectious diseases had a higher likelihood of receiving a dosage adjusted for severe obesity than those who did not. Other factors associated with a high likelihood of adjustment were the number of antimicrobials, BUN between 7 and 10.9 mmol/L, and hemodialysis.

In the univariable analysis, we initially found a significant association between good prescription adjustment (>80%) and the occurrence of unfavourable outcomes. This association is counterintuitive as it implies that optimized dosage leads to negative outcomes. However, the association ceased to exist after adjustment for disease severity and the presence of sepsis on day one. It is common practice to increase antimicrobial dosage in the sickest patients, given their altered antimicrobial pharmacokinetics [32,33]. The presence of severe obesity in these patients is an additional reason to adjust the dosage upwards [5,19]. Finally, the wide therapeutic index of most antimicrobials used in this setting favours adjustments towards higher doses, given the imbalance between the severity of their condition and the low risk of adverse effects associated with overdosing with most antimicrobials. The same pattern was observed when we used adjustment of the first dose or adjustment within the first 72 h of treatment to measure the level of dosage optimization. Interestingly, we found a negative association between secondary outcomes and level of adjustment, but it did not reach statistical significance. 

The literature on the impact of dose adjustment on clinical outcomes in patients with severe obesity treated for infection is scarce. In one study, high doses of tigecycline (100 mg every 12 h) administered to patients with obesity significantly improved clinical outcomes by reducing mortality, ICU stay, recurrent infections, and septic shock events [10]. However, this retrospective cohort study was limited by the small sample (only 11 patients with obesity), and the authors did not adjust for potential confounding factors. In another study, in a subgroup analysis of patients with severe obesity hospitalized for cellulitis, a low antimicrobial dose (TMP-SMX 1 DS PO twice a day or clindamycin 150–300 mg PO every 6–8 h) was associated with a high rate of clinical failure after discharge [7]. Again, this study was limited by its small sample (46 patients with severe obesity) and by the selection of unusual agents for cellulitis treatment [34]. Finally, inadequate dosing but neither weight nor obesity was associated with clinical failure in another study, and patients weighing ≥120 kg were more likely to receive adequate doses of TMP-SMX upon discharge [35]. Time-dependent killing antimicrobials were also associated with worse outcomes but this association did not remain significant after adjustment for covariates. Most patients in our study (*n* = 60, 55%) received both time-dependent and concentration-dependent killing antimicrobials. Since 2010, β-lactams, especially piperacillin-tazobactam have generally been administered as prolonged infusions in the ICU of our center to improve drug exposures. Prolonged perfusion is an important strategy to optimize PD parameters in β-lactams (increasing the time that concentrations remain above the minimum inhibitory concentration (MIC)), rather than only increasing the dose [36,37]. Besides, MIC values and organisms must be considered when assessing effectiveness and outcomes. In our cohort, the impact of bacterial resistance was limited because we excluded episodes where the pathogen was resistant to the antimicrobial received for more than 48 h.

The PBS was chosen to determine the severity of BSI, because it is simple to calculate, and has been described to better predict outcomes in patients with sepsis (which represented 71% of our cohort) than the Acute Physiology and Chronic Health Evaluation II (APACHE II) [17]. Moreover, in retrospective studies, complex scores such as APACHE II or SOFA are likely to be unhelpful due to missing values. We used the modified SOFA (mSOFA) score to limit the impact on missing variables, due to the retrospective nature of our study [22,23].

This study has several limitations. First, the study is subject to biases and missing data due to its retrospective design. Therefore, serum concentrations were not standardized, and it was impossible to perform pharmacokinetic calculations and measure concentrations to assess the appropriateness of the patients’ regimens. Consequently, we decided not to include vancomycin. However, our local guidelines related to adjustments are based on published studies in which PK/PD data were available. We could not assess microbiologic clearance or the duration of BSI because blood samples are not routinely collected after the onset of BSI or they are but at various intervals. To our knowledge, no validated criteria exist to quantify the level of appropriateness. Our classification is subjective, but has been reviewed by two infectious disease specialists (A.C. and L.V.) for its clinical relevance. This classification, although far from being perfect, provides the reader with an order of magnitude regarding adjustment, but further research on this topic is needed. Finally, a posteriori, the study was limited by its small sample size, and it had 38% power to detect a difference of 15% in unfavourable outcomes between patients with and without dose adjustment, coming from a single centre. The small sample size could be explained by the limited proportion of patients hospitalized in our centre with severe obesity and infection treated with antimicrobials requiring adjustment. 

However, despite these limitations, this study is the first to evaluate the association between adjustment for obesity and outcomes in patients with severe obesity and BSIs, such as urinary tract infection, pneumonia, cholangitis, and skin and tissue infections. We could assess short- and medium-term outcomes in several types of infections from various sites and of various severities, from mild symptoms to septic shock, thus providing an overview on the need for and impact of dose adjustment for patients with obesity. Most importantly, each prescription was evaluated considering renal function, which may have changed during hospitalization.

## 5. Conclusions

In conclusion, after adjustment for confounding factors, we did not find an association between dosing optimization and better outcomes in this cohort of patients with severe obesity and BSIs. However, in the absence of measured concentrations of antimicrobials, links between adjusted doses and outcomes can hardly be made. This study was exploratory and ideally a prospective study with the dosage of antimicrobials would be needed. This would maybe allow identifying a link between the adjustment and the outcomes, which we were unable to demonstrate. We did not find any study investigating the link between antimicrobial adjustment and outcomes like we did, in patients with class III obesity hospitalized for a bloodstream infection. Our study is intended as a first step in a field where knowledge remains extremely limited. Yet, given the wide therapeutic index of most antimicrobials and the trend of their effect on secondary outcomes, mortality, and morbidity associated with BSIs, and PK/PD data, it would be wise to continue to adjust antimicrobials upwards in patients with severe obesity and BSIs, while we wait for further evidence. Prolonged infusions also remain important strategies in optimizing PD as they may have a greater influence than dose increment. Finally, we have shown the positive impact of consultations with infectious disease specialists and an antimicrobial stewardship program based on an expert system in increasing the adherence to antimicrobial dosing adapted to patients with obesity.

## Figures and Tables

**Table 1 antibiotics-09-00707-t001:** Dosing regimens for the most frequently prescribed antimicrobials.

Creatinine Clearance *					
Antimicrobial		>50 mL/min	30–50 mL/min	10–30 mL/min	<10 mL/min
**Penicillins**					
ampicillin		2000 mg q4h	2000 mg q6h	2000 mg q6h	2000 mg q6h
penicillin (IV)		4 million units q4h	3 million units q4h	3 million units q4h	2 million units q4h
	(PO)	600 mg q6h	600 mg q6h	600 mg q6h	600 mg q8h
piperacillin/ tazobactam		(CrCl > 40 mL/min) 3000 mg q4h or 4000 mg q6h	(CrCl 20–40 mL/min) 3000 mg q6h	(CrCl 0–20 mL/min) 2000 mg q6h	(CrCl 0–20 mL/min) 2000 mg q6h
**Cephalosporins**					
			(CrCl 35–50mL/min)	(CrCl 10–35 mL/min)	
cefazolin		2000 mg q4h	2000 mg q8h	2000 mg q12h	2000 mg q24h
ceftriaxone		2000 mg q12h	2000 mg q12h	2000 mg q12h	2000 mg q12h
**Quinolones**					
ciprofloxacin (IV)		400 mg q8h	400 mg q12h	400 mg q24h	400 mg q24h
	(PO)	750 mg q12h	500 mg q12h	500 mg q24h	500 mg q24h
**Aminoglycosides**					
gentamicin		1 mg/kg q8h	1 mg/kg q12h	1 mg/kg q24h	1 mg/kg q48h

Abbreviations: CrCl: creatinine clearance, IV: intravenous, PO: oral administration. * estimated with the Cockcroft-Gault equation with adjusted body weight.

**Table 2 antibiotics-09-00707-t002:** Patient demographics and medical conditions stratified by the level of appropriateness.

Characteristics	Good (80–100%) *n* = 47	Moderate (20–79%) *n* = 31	Poor (0–19%) *n* = 32	Total Cohort N = 110
**Female sex**	25 (53)	17 (55)	17 (53)	59 (54)
**Age** (years), median (IQR)	59 (54–66)	66 (51–76)	62 (57–65)	62 (54–67)
**BMI** (kg/m^2^), median (IQR)	45.3 (41.8–50.2)	43.7 (42.2–47.3)	45.0 (42.3–49.2)	44.9 (42.1–48.9)
**Weight** (kg), median (IQR)	127.0 (113.0–145.0)	121.0 (107.5–136.9)	122.0 (108.4–147.9)	124.6 (111.2–142.8)
**Comorbidities**				
Immunosuppression	11 (23)	3 (10)	3 (9)	17 (15)
Coronary artery disease	15 (32)	8 (26)	12 (38)	35 (32)
Diabetes	32 (68)	19 (61)	25 (78)	76 (69)
COPD	10 (21)	9 (29)	8 (25)	27 (25)
Chronic kidney failure	11 (23)	5 (16)	7 (22)	23 (21)
Charlson comorbidity index				
0–3	17 (36)	13 (42)	15 (47)	45 (41)
4–6	26 (55)	9 (29)	9 (28)	44 (40)
≥7	4 (9)	9 (29)	8 (25)	21 (19)
**Infection site**				
Urinary tract	10 (21)	11 (36)	16 (50)	37 (34)
Skin and soft tissue	11 (23)	9 (29)	7 (22)	27 (25)
Pulmonary	8 (17)	4 (13)	1 (3)	13 (12)
Intra-abdominal	2 (4)	3 (10)	1 (3)	6 (5)
Others ^a^	16 (34)	4 (13)	7 (22)	27 (25)
**Severity**				
PBS, median (IQR)	2 (1–4)	1 (0–2)	1 (0–3)	2 (1–3)
PBS ≥ 2	32 (68)	11 (36)	14 (44)	57 (52)
Sepsis on day 1 ^b^	37 (79)	21 (68)	16 (50)	74 (67)
**Pathogens isolated**				
Gram-positive				
*S. aureus*	8 (17)	3 (10)	3 (9)	14 (13)
*S. pneumoniae*	9 (19)	2 (7)	1 (3)	12 (11)
Others ^c^	12 (26)	9 (29)	8 (25)	29 (26)
Gram-negative				
Enterobacteriaceae ^d^	13 (28)	13 (42)	19 (59)	45 (41)
Other ^e^	7 (15)	4 (13)	2 (6)	13 (12)
Polymicrobial infection	5 (11)	3 (10)	2 (6)	10 (9)
**Others**				
Consultation with an infectious disease specialist	31 (66)	13 (42)	10 (31)	54 (49)

Results are reported as number (%) or median (IQR). Abbreviations: IQR: interquartile range, BMI: body mass index, COPD: chronic obstructive pulmonary disease, PBS: Pitt bacteremia score. ^a^ Bones and joints (6), cardiovascular (5), hepatic/biliary (5), catheter (4), undetermined (3), thoracic (2), central nervous system (1), vascular system (1). ^b^ Since it was present upon arrival or occurred early after initiation of antimicrobial treatment, sepsis at day 1 was considered a severity factor rather than a clinical outcome. ^c^ β-hemolytic (groups A, B, C and G) (27) and non-hemolytic streptococci (*S. gallolyticus, S. mitis*) (2). ^d^
*Escherichia coli* (31), *Klebsiella pneumoniae* (10), *Serratia marcescens* (3), *Citrobacter freundii* (2), *Enterobacter cloacae* (2), *Proteus mirabilis* (2), *Morganella morganii* (1), *Klebsiella oxytoca* (1). ^e^
*Enterococcus faecalis* (3), *Haemophilus influenzae* (2), *Aerococcus urinae* (1), *Bacteroides thetaiotaomicron* (1), *Bilophila wadsworthia* (1), *Clostridium septicum* (1), *Clostridium ramosum* (1), *Pasteurella multocida* (1), *Prevotella loescheii* (1), *Pseudomonas aeruginosa* (1).

**Table 3 antibiotics-09-00707-t003:** Clinical outcomes stratified by the level of appropriateness.

Outcomes	Good (80–100%) *n* = 47	Moderate (20–79%) *n* = 31	Poor (0–19%) *n* = 32	Total Cohort N = 110
***Hospital outcomes***				
Time to defervescence (hours), median (IQR)	40.4 (71.9–84.5)	45.2 (17.9–84.5)	53.7 (26.2–90.3)	45 (15.4–87.6)
WBC time to normalization, (hours), median (IQR)	60.6 (24.4–144.8)	70.6 (35.1–111.0)	62.1 (43.8–109.9)	68.3 (32.2–114.0)
Sepsis ^a^	38 (81)	23 (74)	17 (53) *	78 (71)
Day 3	23/46 (50)	16/31 (52)	7/31 (23) *	46/108 (43)
Day 5	17/39 (44)	5/26 (19) *	7/23 (30)	29/88 (33)
Mechanical ventilation	17 (36)	9 (29)	4 (13) *	30 (27)
Duration of mechanical ventilation (days), median (IQR)	4 (2–6)	3 (1–6)	2 (1–7)	3 (2–6)
ICU LOS (hours), median (IQR)	117.6 (67.8–204.7)	107.9 (67.8–141.9)	39.0 (27.4–109.3)	99.7 (43.6–174.5)
LOS (hours), median (IQR)	258.1 (126.7–496.0)	171.9 (117.9–293.3) *	174.6 (98.8–289.1) *	194.5 (114.8–417.4)
***30-day outcomes*** Readmission				
All-causes	4/41 (10)	5/28 (18)	2/32 (6)	11/101 (11)
Relapse	1/41 (2)	3/28 (11)	1/32 (6)	5/101 (5)
Time to readmission (days), median (IQR) ^b^	13 (3–21)	11 (5–19)	-	11 (7–18)
All-cause 30-day mortality	6 (13)	3 (10)	0	9 (8)

Results are reported as number (%) or median (IQR). Abbreviations: IQR: interquartile range, WBC: white blood cells, ICU: intensive care unit, LOS: length of stay. ^a^ At least one day with mSOFA ≥ 2, on days 1, 3, 5, 7. ^b^ In cases with poor appropriateness, only 2 patients were readmitted with time to readmission of 9 and 18 days. * Statistically significant difference (*p* value < 0.05), reference category: good (80–100%).

**Table 4 antibiotics-09-00707-t004:** Factors associated with adjusted antimicrobial therapy.

Factors	No. Adjusted Therapy/ Total (%)	Univariable OR (95% IC)	*p* Value	Multivariable OR (95% IC)	*p* Value
**Number of antimicrobials**	-	2.19 (1.43–3.14)	<0.001	2.17 (1.40–3.37)	<0.001
**Consultation with an infectious disease specialist**					
No	16/56 (29)	reference		reference	
Yes	31/54 (57)	3.37 (1.53–7.44)	0.003	3.33 (1.29–8.58)	0.013
**Sepsis on day 1**					
No	10/36 (28)	reference			
Yes	37/74 (50)	2.60 (1.10–6.14)	0.03		
**BUN (mmol/L)**					
<7	7/32 (22)	reference		reference	
7–10.9	14/22 (64)	6.25 (1.87–20.90)	0.003	7.34 (1.83–29.48)	0.005
≥11	25/52 (48)	3.31 (1.22–8.98)	0.02	2.51 (0.74–8.46)	0.14
missing	1/4 (25)	1.190 (0.11–13.30)	0.89	0.51 (0.03–8.49)	0.6
**APSS**					
No	17/51 (33)	reference			
Yes	30/59 (51)	2.07 (0.95–4.49)	0.066		
**Immunosuppression**					
No	36/93 (39)	reference			
Yes	11/17 (65)	2.90 (0.99–8.54)	0.05		
**Hemodialysis**					
No	39/100 (39)	reference		reference	
Yes	8/10 (80)	6.26 (1.26–31.01)	0.03	10.30 (1.62–65.56)	0.014
**Charlson comorbidity index**					
0–3	17/45 (38)	reference			
4–6	26/44 (59)	2.38 (1.02–5.57)	0.05		
≥7	4/21 (19)	0.388 (0.11–1.35)	0.14		
**PBS**					
0–1	15/53 (28)	reference			
≥2	32/57 (56)	3.24 (1.47–7.18)	0.004		
**Infection site**					
Urinary	10/37 (27)	reference			
Pulmonary	8/13 (62)	4.32 (1.14–16.37)	0.03		
Skin and soft tissue	11/27 (41)	1.86 (0.65–5.34)	0.25		
Other	18/33 (55)	3.24 (1.19–8.79)	0.02		

Reference category: 0–19% poor (vs. 20–100%). Abbreviations: BUN: blood urea nitrogen, APSS: Antimicrobial Prescription Surveillance System, PBS: Pitt bacteremia score.

**Table 5 antibiotics-09-00707-t005:** Factors associated with unfavourable outcomes.

Factors	No. Unfavourable Outcomes/ Total (%)	Univariable OR (95% CI)	*p* Value	Multivariable OR (95% CI)	*p* Value
**Age** (years)	-	1.05 (1.01–1.08)	0.01	1.07 (1.02–1.12)	0.009
**BMI** (kg/m^2^)	-	1.02 (0.95–1.08)	0.64		
**Charlson comorbidity index**	-	1.19 (1.02–1.38)	0.02		
**Hemodialysis**					
No	58/100 (58)	reference			
Yes	2/10 (20)	0.18 (0.04–0.90)	0.04		
**PBS ≥ 2**					
No	15/53 (28)	reference		reference	
Yes	45/57 (79)	9.50 (3.97–22.75)	<0.001	7.30 (2.09–25.52)	0.002
**Sepsis on day 1**					
No	4/36 (11)	reference		reference	
Yes	56/74 (76)	24.89 (7.75–79.97)	<0.001	16.78 (3.93–71.63)	<0.001
**Infection site**					
Urinary	11/37 (30)	reference		reference	
Pulmonary	10/13 (77)	7.88 (1.81–34.28)	0.01	7.52 (1.20–47.15)	0.031
Skin and soft tissue	20/27 (74)	6.75 (2.22–20.55)	0.001	7.79 (1.67–36.41)	0.009
Other	19/33 (58)	3.21 (1.20–8.60)	0.02	9.47 (1.99–45.10)	0.005
**BUN (mmol/L)**					
<7	9/32 (28)	reference			
7–10.9	16/22 (73)	6.82 (2.02–22.95)	0.002		
≥11	35/52 (67)	5.26 (2.01–13.80)	0.001		
missing	0/4 (0)	-			
**Appropriateness category**					
Good (80–100%)	30/47 (64)	2.94 (1.16–7.46)	0.02		
Moderate (20–79%)	18/31 (58)	2.31 (0.84–6.34)	0.11		
Poor (0–19%)	12/32 (38)	reference			
**Number of antimicrobials**	-	1.98 (1.34–2.93)	0.001		
**Type of antimicrobial** (based on PD)					
Concentration- dependent	1/11 (9)	reference			
Time- dependent	25/38 (66)	19.23 (2.21–167.11)	0.007		
Mixed	34/61 (56)	12.59 (1.52–104.58)	0.02		

Abbreviations: BMI: body mass index, PBS: Pitt bacteremia score, BUN: blood urea nitrogen, PD: pharmacodynamics.

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
