# Peer review of "Is Antimicrobial Dosing Adjustment Associated with Better Outcomes in Patients with Severe Obesity and Bloodstream Infections? An Exploratory Study"

_antibiotics, 2020, doi:10.3390/antibiotics9100707_

Round 1
Reviewer 1 Report
The Authors carefully analyzed single center's data about the relationship of category III obesity and antibiotic dose adjustment and outcome. They concluded that the dose adjustment is not an independent predictor of pure outcome.
There are only two comments:
- The final version of paper must follow the conservative sequence of publication: The Methods must be moved before the Results to improve the understanding.
- One theroretical point should be elucidated. The results are suggesting that the severe of patients condition (BUN, hemodialysis...), the better the dose adjustment. Thus, the good appropriateness category is paradoxically connected to pure prognosis (OR: 2.94). These result probably suggests that severe patients were admitted to ICU leading to carefully monitoring and appropriate antibiotic treatment. This question have to be clarified before publicaton (patient admission to different wards, its relation to antibiotic dose adjustment) . Otherwise the readers may conclude false facts.
Author Response
- The final version of paper must follow the conservative sequence of publication: The Methods must be moved before the Results to improve the understanding.
We moved the Methods before the Results and reference numbering has been changed accordingly.
- One theroretical point should be elucidated. The results are suggesting that the severe of patients condition (BUN, hemodialysis...), the better the dose adjustment. Thus, the good appropriateness category is paradoxically connected to pure prognosis (OR: 2.94). These result probably suggests that severe patients were admitted to ICU leading to carefully monitoring and appropriate antibiotic treatment. This question have to be clarified before publicaton (patient admission to different wards, its relation to antibiotic dose adjustment) . Otherwise the readers may conclude false facts.
To better clarify this point, we added the following sentences (Results, page 7, lines 215-221).
Overall, 53 % (n = 58) of patients were admitted to the ICU and the median time to ICU admission was 7.4 hours (IQR 4.0-14.9). Patients in the good appropriateness category tended to be admitted sooner (5.6 hours IQR 3.7-10.4, p = 0.25) compared with the other groups (9.4 hours [IQR 5.0-25.4]; 8.0 hours [IQR 4.0-21.8]. Time from admission to first effective antimicrobial did not differ between groups (p = 0.84). The first antimicrobial was administered before ICU admission in most patients (89%, n = 98), but half of patients who received their first antimicrobial in the ICU were in the good level of appropriateness. In the original discussion, this counterintuitive association was already mentioned in lines 181 to 190 (revised discussion, lines 289-298).
We also added the definition of the time from admission to first effective antimicrobial in Materials and Methods (page 3, lines 90 to 92) : The time to effective antimicrobial was determined by the time between admission and the administration of the first effective antimicrobial related to the infection.
Reviewer 2 Report
I read with interest the manuscript of Sirard et al “Antimicrobial dosing adjustment is not associated with better outcomes in patients with severe obesity and bloodstream infections”
This manuscript assess factors associated with adjustment of antimicrobial dosing and compares clinical outcome according to the appropriateness of antimicrobial dose adjustments.
I have some comments that might help to improve this manuscript:
- Results, page 2, line 62. The classification of the adequacy of the treatment based on the percentage of similarity with the theoretical dose is not validated. This is an inconvenience since it is the criterion that has been used to classify patients. The authors should comment on this aspect.
- Results, page 4, line 102. Authors should explain what the APSS consists of and how it works
- Results, outcomes. Page 4, line 11. In my opinion outcomes as the time to defervescence and WBC time to normalization, have little clinical relevance and may be influenced by other factors.
- Results, page 5, line 134. Consultation with an infectious Disease specialist is not generalizable. For example, in ICUs, the intensivists themselves are the experts in infectious diseases in critical patients and it's not necessary to resort to the clinical infectologist.
- Discussion, page 10, line 226. The authors mention that was impossible to perform pharmacokinetic calculations and measure concentrations of antimicrobials. I think that this limitation completely invalidates the study.
- Materials and methods, page 10. It would be interesting if the authors provided the total number of patients with grade III obesity who were admitted during the study period (also include it in the flowchart)
Author Response
- Results, page 2, line 62. The classification of the adequacy of the treatment based on the percentage of similarity with the theoretical dose is not validated. This is an inconvenience since it is the criterion that has been used to classify patients. The authors should comment on this aspect.
We added to the discussion the following section (page 13, lines 339 to 343):
To our knowledge, no validated criteria exist to quantify the level of appropriateness. Our classification is subjective, but has been reviewed by two infectious disease specialists (A.C. and L.V.) for its clinical relevance. This classification, although far from being perfect, provides the reader with an order of magnitude regarding adjustment, but further research on this topic is needed.
- Results, page 4, line 102. Authors should explain what the APSS consists of and how it works
We added the following description (Materials and Methods, page 4, lines 139 to 143):
The Antimicrobial Prescription Surveillance System (APSS) is an asynchronous system that generates alerts for potentially inappropriate antimicrobial prescriptions based on published recommendations and expert opinions. These alerts are reviewed by pharmacists who are part of the antimicrobial stewardship program team and recommendations are made to physicians. Special rules were developed for patients with class III obesity.
- Results, outcomes. Page 4, line 11. In my opinion outcomes as the time to defervescence and WBC time to normalization, have little clinical relevance and may be influenced by other factors.
We agree with the reviewer’s comment. We removed lines 213 to 215 and 301 to 304 where a mention was made on these variables. However, we still think these exploratory secondary outcomes have some clinical relevance and decided to keep them in table 3.
- Results, page 5, line 134. Consultation with an infectious Disease specialist is not generalizable. For example, in ICUs, the intensivists themselves are the experts in infectious diseases in critical patients and it's not necessary to resort to the clinical infectologist.
We do not agree with this comment, this is not the case in our center. We think that our experience is representative of the Canadian practice but may not apply in other centers, worldwide.
- Discussion, page 10, line 226. The authors mention that was impossible to perform pharmacokinetic calculations and measure concentrations of antimicrobials. I think that this limitation completely invalidates the study.
We acknowledge that this is a limitation of our study and thus it reduces the conclusions that we can draw from it.
We added the following sentence (Discussion, page 13, lines 336 to 338):
However, our local guidelines related to adjustments are based on published studies in which PK/PD data were available.
We also added the following sentences to put emphasis on this very important comment (Conclusion, page 13, lines 360 to 366).
However, in the absence of measured concentrations of antimicrobials, links between adjusted doses and outcomes can hardly be made. This study was exploratory and ideally a prospective study with the dosage of antimicrobials would be needed. This would maybe allow identifying a link between the adjustment and the outcomes, which we were unable to demonstrate. We did not find any study investigating the link between antimicrobial adjustment and outcomes like we did, in patients with class III obesity hospitalized for a bloodstream infection. Our study is intended as a first step in a field where knowledge remains extremely limited.
- Materials and methods, page 10. It would be interesting if the authors provided the total number of patients with grade III obesity who were admitted during the study period (also include it in the flowchart)
Unfortunately, we do not have this total number of patients because we precisely identified patients from a list of patients with a BMI > 40 and hospitalized for a bloodstream infection. However, flowchart of included and excluded episodes is provided in Supplementary Materials (Figure S1).
Round 2
Reviewer 1 Report
The Authors carried out all the needed corrections.
Author Response
Based on the reviewer's comments, no changes are required.
Reviewer 2 Report
The authors have responded adequately to my queries
Author Response

(The authors gave the same response as above.)
